# Mechanical Behavior of Single Patch Composite Repaired Al Alloy Plates: Experimental and Numerical Analysis

**DOI:** 10.3390/ma13122740

**Published:** 2020-06-17

**Authors:** Jingtao Dai, Peizhong Zhao, Hongbo Su, Yubo Wang

**Affiliations:** Department of aeronautical and mechanical engineering, Naval Aeronautical and Astronautical University Qingdao Branch; Qingdao 266041, China; zpzgraduate@163.com (P.Z.); woshi_asu@163.com (H.S.); xwyk1@126.com (Y.W.)

**Keywords:** GFRP composite repair, tensile strength, failure mechanism, finite element model

## Abstract

In this paper, glass fiber reinforced polymer (GFRP) materials were used to repair cracked Al plates. In order to study the influences of resin properties and repair configurations, three resins and two patch configurations were selected to manufacture six groups of specimens. It turned out that only little differences (less than 3%) were found in tensile strength among the six groups. Compared with the parent plates, the strength recovery ratio was higher than 80% after the GFRP repair, representing excellent repair efficiency. Moreover, a finite element model (FEM) was established to analyze the failure process of the repaired structure under tensile loading. The FEM results show good agreement with the experimental results, indicating good precision. Both the experimental and numerical work found that the damage initiated in the plies adjacent to the crack surface and the failure modes was mainly delamination and fiber breakage. This work will be meaningful for the future application of GFRP in metallic structures.

## 1. Introduction

Nowadays, fiber reinforced polymer (FRP) materials were widely used to strengthen or repair many different kinds of structures, including steel reinforced concrete in civil engineering [1,2,3,4], oil and gas transmission/transportation pipelines [5,6,7], aircraft metal structures [8,9] and also composite aircraft structures [10,11,12]. Compared with a traditional mechanical fasten repair, the FRP repair technique provides lower stress concentration, lightweight, excellent corrosion resistance, lower cost, easier operation and higher repair efficiency. Moreover, continuous fiber reinforced polymer composites possess improved strength-to-weight ratios and designability. From the above research work, it was worth noting that there will be prosperity and development in FRP composite repair techniques.

Many different kinds of FRP composite repaired structures were investigated to acquire their static strength, fatigue strength, durability and so on. A large number of experimental tests and numerical models were conducted. Mohsen et al. [13] reviewed the environmental durability of adhesively bonded FRP/steel joints in civil engineering. Moisture and temperature are identified as the most critical environmental factors resulting in complex failure mechanisms. Rohem et al. [7] investigated the performances of GFRP composite repaired steel pipes. Two types of defects, i.e., through wall and non-through wall, in steel pipes, were chosen and good satisfactory was obtained for both. Failure occurs at the laminate-substrate interface for through-wall defects, while no failure occurs in the composite for non-through wall defect. Fujimoto et al. [14] identified the locations and shapes of crack and disbond fronts of the aircraft structural panel repaired with bonded FRP composite patches through an analytical model. Measured strain data has verified the model’s validity. Tao Chen et al. [15] studied the fatigue crack growth life of cruciform steel welded joints repaired with FRP materials. A numerical model was built, and the adhesive used to bond the composites with the steel plate was modeled by interface spring elements between the composite patches and the steel plate. The crack growth rate was correlated with the amplitude of the applied stress intensity factor (SIF). It showed that the FRP patch can reduce the crack growth rate and crack opening displacement. Benyahia F et al. [16] investigated the aging effects on a repaired aluminum alloy 7075 T6 plate. It showed that fatigue life can be increased by fifteen times by the patch. A three-dimensional virtual crack closure technique (3D VCCT) was used to calculate the SIF. Results showed that the SIF reduced 80% by the composite patch. In addition, it was found that an aged composite patch can reduce the SIF further. Albedah et al. [17] studied the effects of patch length on fatigue life of the repaired plate in both the 2024 T3 and 7075 T6 aluminum alloy. It showed that the fatigue life increases with the patch length. The SIF at the patched surface reduces with patch length, while SIF at the free surface increases with the patch length. Ouinas et al. [18] investigated the effects of the disbond area on SIF of aluminum panels repaired with boron composite patch. In addition, the effects of the orientation, thickness and width of the patch were also discussed. Results showed that setting fiber direction along the crack can reduce the SIF. The SIF increases with the increase of the disbonds. Shafique et al. [19] carried out research on the influences of CFRP repair patch on the fatigue properties of steel plates. The results showed that fatigue life increased significantly, especially for the combination usage of drilling a conventional crack-stop hole with CFRP composite repair. Sotirios et al. [12] employed thermography to investigate the patch debonding propagation subject to cycling mechanical loading, which demonstrates both quantitative and qualitative information for repair efficiency. Pradhan et al. [20] investigated the mechanical properties of edge-cracked aluminum plates repaired with a woven glass/epoxy composite patch. Tensile tests, flexural tests and Rockwell hardness tests were conducted, and effects of patch thickness and orientation were discussed. Maleki et al. [21] used acoustic emission and fractography images to investigate fatigue properties of single-sided composite patch repaired aluminum 6061 alloy plates. It was found that eight layers of the patch improved fatigue life about 197% compared with the unrepaired. Effects of plasticity on the composite patched central cracked aluminum plate were investigated by Hart and Bruck [22], which indicates that the patched specimen has a much larger area with high plastic strain under the same crack opening displacement (COD). Bouzitouna et al. [23] combined the composite patch repair technique and hole drilling technique, which showed an increase in mechanical strength and fatigue life.

In this article, aluminum alloy structures applied in aircraft were chosen as to-be-repaired objects. As for aircraft manufactured in the last three decades, there is an urgent need for extending their service life, especially for navy aircraft. The FRP composite repair technique, as an efficient and cost-effective method, would play an important role. However, more data and clearer cognition on mechanical behavior are needed for better engineering applications. In order to realize an economic and practical maintenance solution, a woven fabric made of glass fiber and a room temperature curing resin system was used to repair Al plates. Specimens were manufactured by laying up wet GFRP prepregs onto the cracked Al plates. Tensile tests were carried out to evaluate the repair efficiency. Failure mechanisms were analyzed by observation of fracture surfaces. A numerical model was also established for a better understanding of its mechanical behavior.

## 2. Experimental Details

### 2.1. Material and Specimen Preparation

The repair efficiency of three material systems was compared in the present work: SW90-a/EA9396, SW90-a/EA9394, SW90-a/EA9396+EA9394. Three different resins were selected to compare their characteristics in practical repair process and repair efficiency, so as to choose the easiest repair process. The properties of the two epoxy resins EA9394 and EA9396 are listed in Table 1. EA9394 had better mechanical properties, but its viscosity was much higher leading to poor handleability. While EA9396 resin had a relatively low viscosity and good infiltration, which was beneficial for the hand lay-up repair process. Moreover, these two resins can be cured under room temperature. To simulate the service damages, we manufactured pre-fabricated cracks in rectangle aluminum LY12CZ plates using the laser cutting equipment. The LY12CZ is a kind of common aeronautical engineering Al alloy. The thickness of the Al plate was 1.5 mm. The crack size was 10 mm (length) × 0.3 mm (width), as shown in Figure 1. Before the repair procedure, the surface of the Al plates was abraded and treated with two kinds of silane coupling agents: AC130 and KH550. Afterward, wet prepregs were patched to the surface ply by ply according to the following configurations of patches (Figure 2). Six plies were laid in two ways: triangle and inverted triangle. Each layer was 0.1mm after cured and the overall thickness of the patch was approximately 0.6 mm. The width of prepregs was the same as the Al plate, which was 50 mm. Lengths of the prepregs were 62 mm, 72 mm, 82 mm, 92 mm, 102 mm and 112 mm for each ply. The selection of the shortest length of the prepreg aimed to reduce the stress concentration around the crack tips. The specimens were cured in vacuum bags at room temperature for 48 h and post-cured for more than 3 days before testing.

### 2.2. Test Procedures

To evaluate the repair efficiency, we performed tensile tests on a material testing machine TST-DL4205 with a loading capacity of 10 tons. The schematic of specimens and tensile testing instruments is shown in Figure 3. The cross-head speed was 1 mm/min and a strain data acquisition system was used to monitor deformation in critical positions.

### 2.3. Results and Discussion

#### 2.3.1. Tensile Strength

Figure 4 shows typical load-displacement curves (*P*-*δ* curves) of GFRP-repaired cracked plates under unidirectional tensile loads. After the reparation with the wet GFRP prepregs, the cracked plate recovered strength and structural modulus to some extent. However, the plastic mechanical characteristic is lost. The type of prepregs with different resin materials had little influence on the repair efficiency. Table 2 summarizes the experimental data for the tensile strength and restoration ratio for all specimens. The tensile strength of the specimen was calculated by dividing the ultimate load by cross-sectional area of the specimen. The load capacity restoration ratio was calculated by dividing the ultimate tensile load of the repaired plate with that of the original plate. The specimens with triangle patches had a slightly higher restoration ratio than those with inverted-triangle patches. The mixture usage of two different resins exhibited the highest tensile strength, while the specimen with EA9396 resin showed the lowest tensile strength. However, the differences between these three groups were very small, less than 3%. The load capacity recovery ratio was higher than 80% after the GFRP repair, representing excellent repair efficiency.

#### 2.3.2. Damage Observation in Specimens

As can be seen in Figure 5, the damage to the sample under axial tensile load can be visually observed at the crack when the load increased to 18 kN. Afterward, the damage extended around the crack with a further increase in the load. New damaged places appeared around the corner of the patch at a load of 27 kN and the final fracture happened at approximately 29 kN.

The visual inspection of fracture surfaces is shown in Figure 6. A neat fracture along the crack of the Al plate can be observed. The damage modes of the patch were mainly the fracture of adhesively bonded joints and adhesive-composite interface damage. Fiber breakage occurred subsequently after adhesive failure. The fracture area highlighted by the red box in Figure 6 was examined through scanning electronic micrograph. From the scanning electronic micrograph shown in Figure 7, adhesive failure of the adhesively bonded composite joint can be seen.

## 3. Finite Element (FE) Model Analysis

### 3.1. Finite Element Model

A three-dimensional finite element model was built in the commercial finite element analysis software Abaqus 6.11. Hexahedral elements C3D8R were selected to represent the behavior of both composites and Al materials in this model. As presented in Figure 8, green elements denote composite materials, while grey elements illustrate Al. To simulate the GFRP patch, there was one element in each layer and the material orientation was defined. The area near the crack was meshed with much smaller elements with sizes of 0.06 mm × 0.2 mm × 0.3 mm for Al and 0.06 mm × 0.1 mm × 0.3 mm for the composite. The left edge was fixed and the axial tensile load was applied on the right edge. Coupling constraints were applied on the edges to reference points RP1 and RP2, respectively.

### 3.2. Damage Criteria and Progressive Damage Model

To predict the loading capacity and the damage modes of the GFRP-repaired Al plates, both the failure mechanism of Al and the composite were considered. The ductile mechanical behavior of the Al plate was simulated using a traditional metal ductile model and the orthotropic mechanical behavior and damage procedure of the patch were simulated through implanting a USDFLD subroutine. A modified failure model based on 3D stresses [24] in a composite layer with progressive failure modeling capability was established for each plain-weaved glass fabric layer. In this study, seven failure modes were evaluated and the damage initiation rules were list as follows:

in-plane fill direction tensile failure ($\sigma_{22}\geq 0$):(1)$${\left(\frac{\sigma_{22}}{Y_{T}}\right)}^{2}+{\left(\frac{\sigma_{23}}{S_{23}}\right)}^{2}\geq 1;$$
in-plane fill direction compression failure ($\sigma_{22}<0$):(2)$${\left(\frac{\sigma_{22}}{Y_{C}}\right)}^{2}\geq 1;$$
in-plane warp direction tensile failure ($\sigma_{11}\geq 0$):(3)$${\left(\frac{\sigma_{11}}{X_{T}}\right)}^{2}+{\left(\frac{\sigma_{13}}{S_{13}}\right)}^{2}\geq 1;$$
warp yarn compressive failure ($\sigma_{11}<0$):(4)$${\left(\frac{\sigma_{11}}{X_{C}}\right)}^{2}\geq 1;$$
in-plane shear-out failure:(5)$${\left(\frac{\sigma_{12}}{S_{12}}\right)}^{2}\geq 1;$$
out-of-plane delamination in tension ($\sigma_{33}\geq 0$):(6)$${\left(\frac{\sigma_{33}}{Z_{T}}\right)}^{2}+{\left(\frac{\sigma_{13}}{S_{13}}\right)}^{2}+{\left(\frac{\sigma_{23}}{S_{23}}\right)}^{2}\geq 1;$$
out-of-plane delamination in compression ($\sigma_{33}<0$):(7)$${\left(\frac{\sigma_{33}}{Z_{C}}\right)}^{2}+{\left(\frac{\sigma_{13}}{S_{13}}\right)}^{2}+{\left(\frac{\sigma_{23}}{S_{23}}\right)}^{2}\geq 1.$$

The mechanical properties of the plain-weaved glass fabric composite layer and the Al plate used in the model are shown in Table 3. When damage predicted by the above criterion occurred within elements, a set of degrading rules was used to relate the damage growth with the stiffness loss of the material. Table 4 shows the stiffness degrading ratio according to each damage mode. For the sake of numerical convergence, when the mechanical property was degraded to zero, it was replaced by a very small value of 0.01.

### 3.3. Explaining Damage Mechanism by Using Simulations

Load-displacement curves obtained through the model were compared with experimental results as shown in Figure 9. It indicated that the structural stiffness and load capacity were restored almost 80%~90% of the original plate. The boundary condition and the material failure model the FEA model can hardly be the same as those of the practical tests. It may lead to the differences between the FEA results and test results. Meanwhile, the strains from the model were also investigated and a good correlation was found with experimental results, see in Figure 10, which suggests a relatively high precision of the model. Strains 2 and 4 were on the patch side, while strains 1 and 3 were on the back side. It showed that strain on the patch side was smaller, which indicated that the repaired plate was bending towards the back side.

As can be seen in Figure 11 and Figure 12, stress concentration around the tips decreased due to the addition of the patch. The stress near the patch side (top view) was significantly lower than on the back of the Al plate (bottom view). At 80% of the ultimate load, the tip of the Al plate came to the plastic yielding stage and crack propagates until the final ductile fracture.

The most dominant damage modes of the GFRP patch were delamination and fiber breakage in the warp yarn direction, shown in Figure 13. Delamination in compression occurred first in the patch adjacent to the Al plate around the crack and then propagated along the length direction. Fiber breakage in the yarn direction occurred later and it was mainly located in the two edges of the patch adjacent to the Al plate.

## 4. Conclusions

In this study, both experimental tests and numerical analysis were performed to investigate the tensile strength and failure mechanism of the GFRP-repaired Al plate, and the results can be summarized as follows:

1. We compared three different resins used for prepregs, and no significant difference was found, being less than 3%. Moreover, two different patch configurations showed little influence in the ultimate strength.

2. The GFRP repair’s strength recovery ratio was found to be higher than 80% of the original Al plate, which represents an excellent repair efficiency for a single strap repair.

3. The FE model exhibited good accuracy in predicting the load-displacement response and the predicted failure modes were also consistent with experimental observation.

4. Both the experimental and numerical results illustrate that the damage initiated around the crack tip and was mainly adhesive bond failure and interfacial failure between the patch and Al plate. At the ultimate load, final ductile failure in the Al plate and fiber breakage in the warp yarn direction occurred.

## Figures and Tables

**Figure 1 materials-13-02740-f001:**
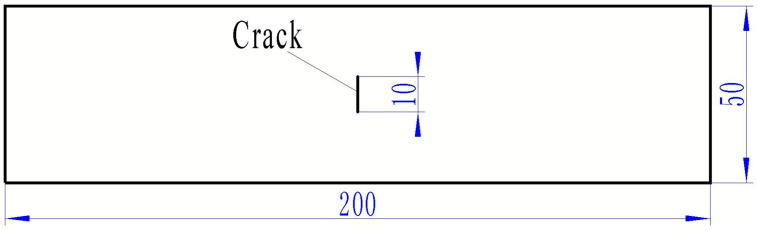
Geometrical configuration of Al plate with a generated crack. (Unit: mm).

**Figure 2 materials-13-02740-f002:**
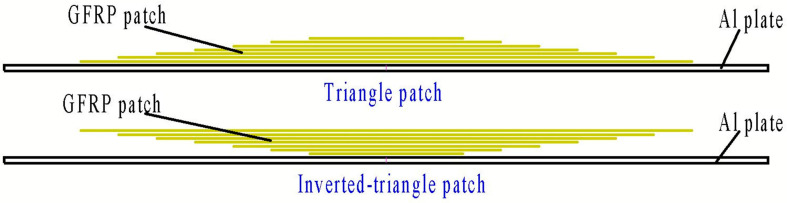
The repair configuration of the single-sided external patch.

**Figure 3 materials-13-02740-f003:**
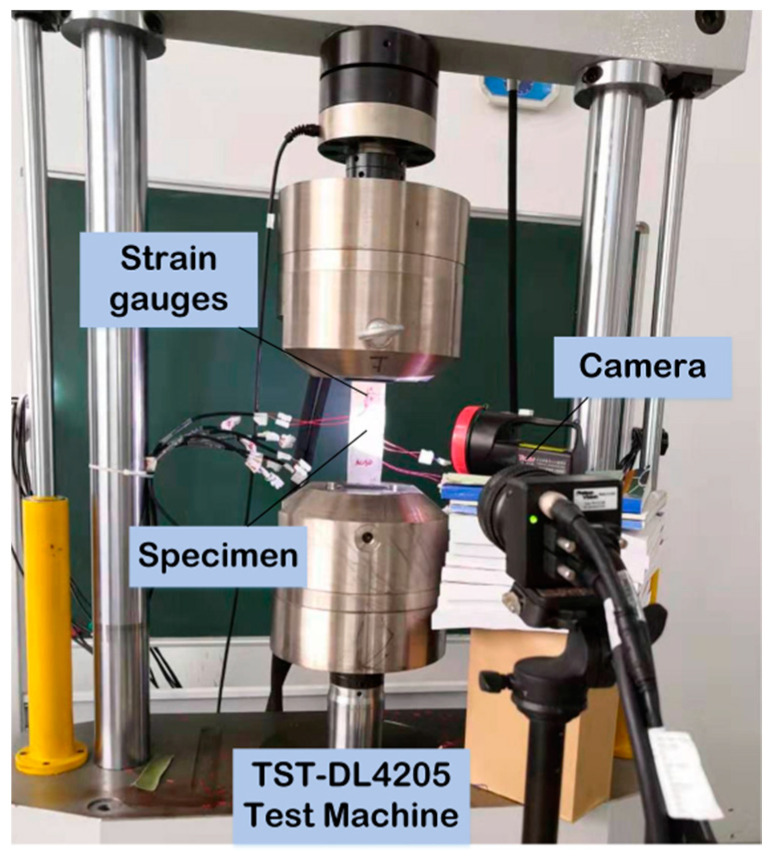
The schematic representation of the specimen and the set-up for tensile strength testing.

**Figure 4 materials-13-02740-f004:**
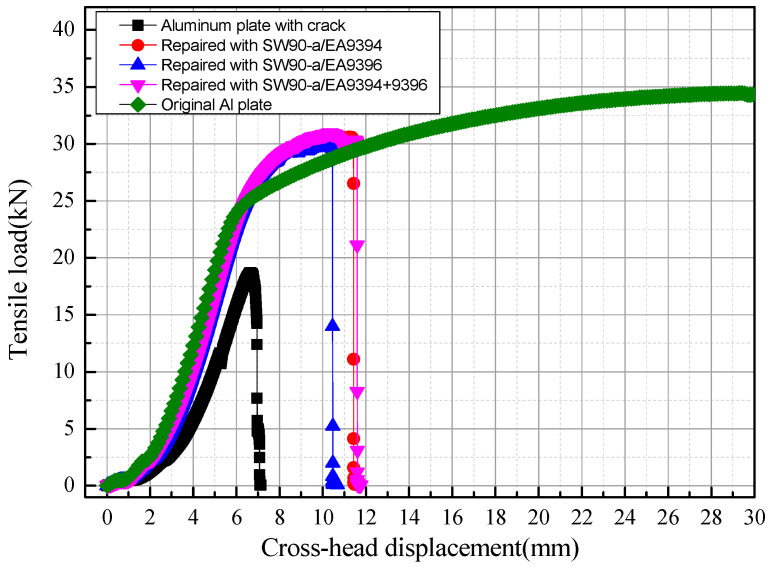
Typical tensile load-displacement curves of the GFRP-repaired cracked plates.

**Figure 5 materials-13-02740-f005:**
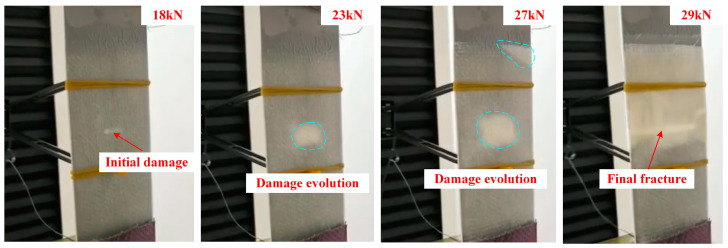
Visual inspected damage process during loading.

**Figure 6 materials-13-02740-f006:**
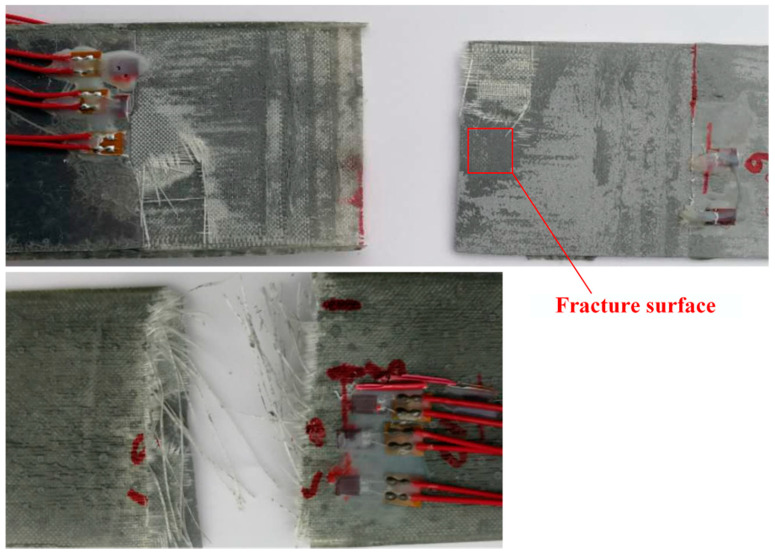
The visual examination of the fracture surface after the testing.

**Figure 7 materials-13-02740-f007:**
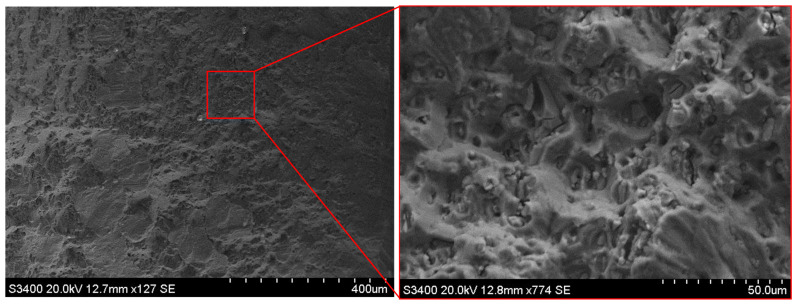
The scanning electronic micrograph of the adhesively bonded composite joint’s fracture surface.

**Figure 8 materials-13-02740-f008:**
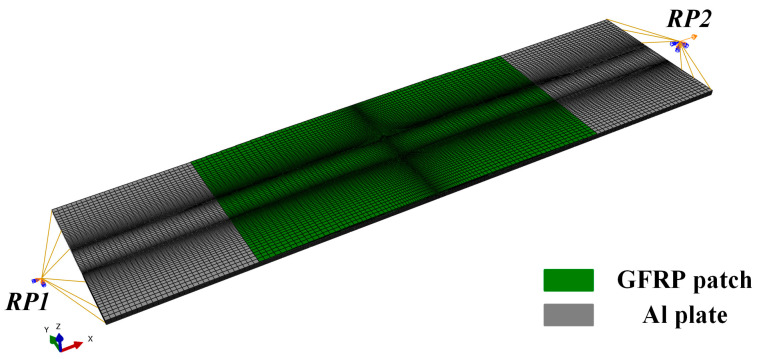
Finite element model for the GFRP-repaired Al plate with center-crack.

**Figure 9 materials-13-02740-f009:**
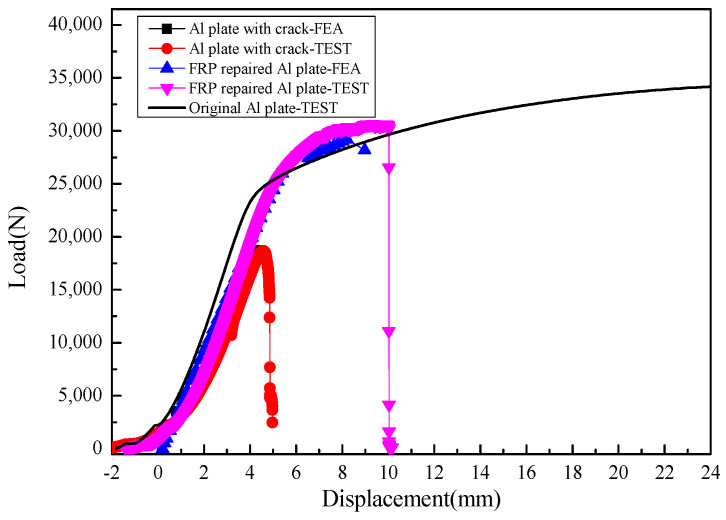
The comparison of load-displacement curves obtained from FEM and experiments.

**Figure 10 materials-13-02740-f010:**
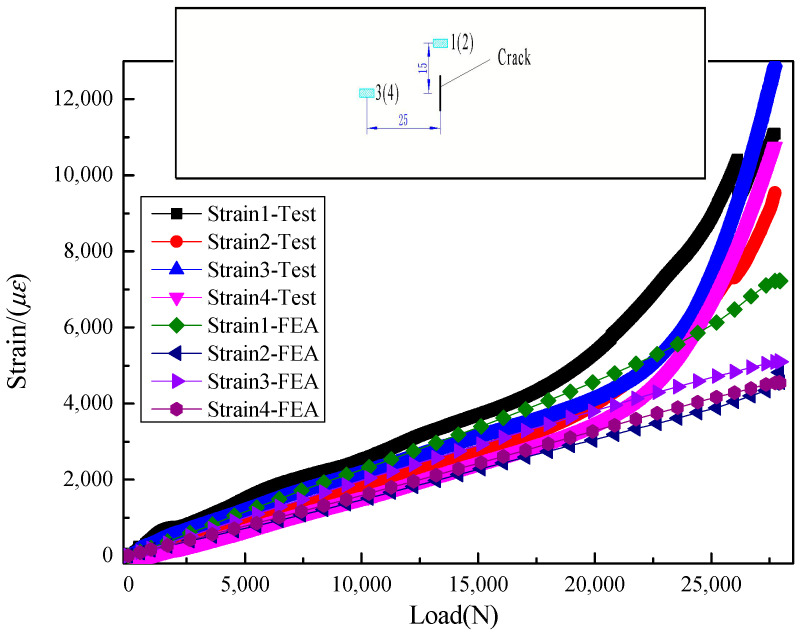
The comparison of strain-load curves obtained from FEM and experiments.

**Figure 11 materials-13-02740-f011:**
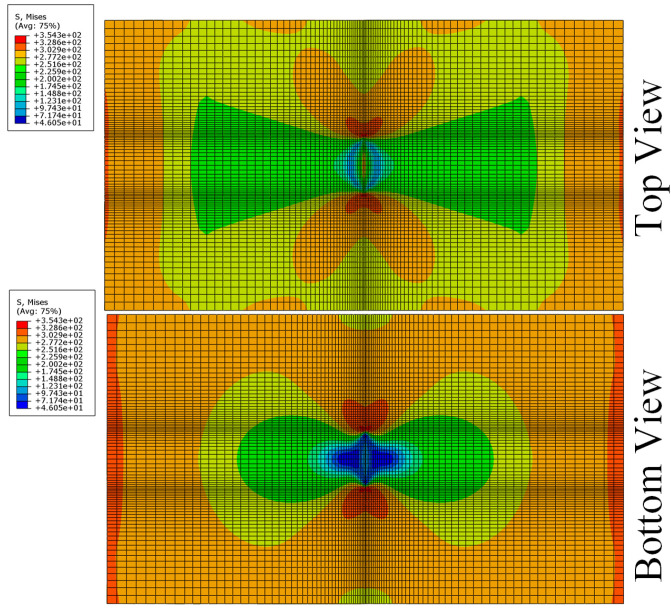
The contour of Mises stress in the Al plate under tensile loading.

**Figure 12 materials-13-02740-f012:**
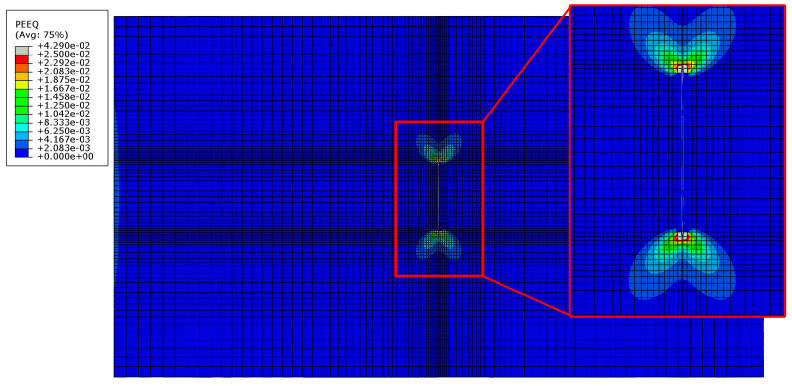
The plastic strain around the crack tip of the Al plate.

**Figure 13 materials-13-02740-f013:**
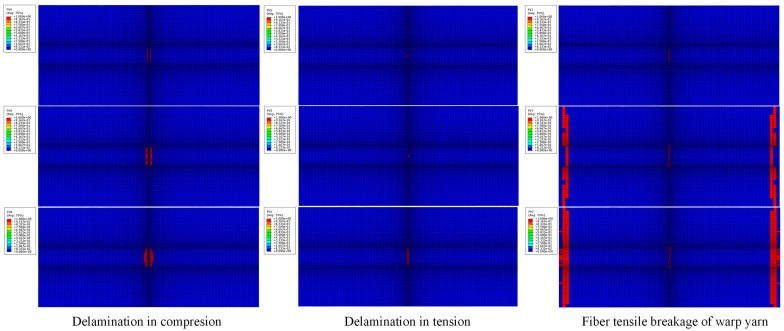
Failure modes and damage propagation in the GFRP patch.

**Table 1 materials-13-02740-t001:** The properties of the two kinds of resin at room temperature (25 °C).

Resin Type	EA9396	EA9394
Tensile strength (MPa)	35.2	46
Tensile modulus (GPa)	2.750	4.237
Elongation at tensile failure (%)	3.4	1.66
Shear strength (MPa)	27.6	28.9
Shear modulus (MPa)	948.3	1461
Viscosity (Pa·s)	~3.5	~450

**Table 2 materials-13-02740-t002:** Tensile strength and repair recovery ratio of the GFRP-repaired plates.

Patch Material	Specimens with Triangle Patch			Specimens with Inverted Triangle Patch		
	Ultimate Failure Load (kN)	Tesile Strength (Mpa)	Recovery Ratio (%)	Ultimate Failure Load (kN)	Tesile Strength (Mpa)	Recovery Ratio (%)
SW90-a/EA9394	28.59 28.73 28.66	272.29 273.62 272.95	83.77 84.17 83.97	28.36 28.45 28.56	270.10 270.95 272.00	83.09 83.36 83.68
Average	28.66	272.95	83.97	28.46	271.05	83.39
SW90-a/EA9396	28.01 28.35 27.97	266.76 270.00 266.38	82.07 83.06 81.95	27.92 27.32 28.62	265.90 260.90 272.57	81.80 80.05 83.86
Average	28.11	267.71	82.36	27.95	266.19	81.89
SW90-a/EA9394+9396	28.70 29.08 29.15	273.33 276.95 277.62	84.09 85.20 85.41	28.14 28.86 28.49	268.00 274.86 271.33	82.45 84.56 83.47
Average	28.98	276.00	84.91	28.50	271.43	83.50

Note: The ultimate failure load and tensile strength of the original Al plate and the unrepaired Al plate with crack are 34.13 kN, 455.07 MPa and 18.68 kN, 249.07 MPa, respectively. Compared with the original Al plate, the residual load capacity ratio of the cracked Al plate is 54.73%.

**Table 3 materials-13-02740-t003:** Mechanical properties of SW90-a/EA9394 lamina.

Plain Weaved Glass Fabric Lamina Reinforced EA9394 Resin								
E_11_ (GPa)	E_22_ (GPa)	E_33_ (GPa)	G_12_ (GPa)	G_23_ (GPa)	G_13_ (GPa)	ν_12_	ν_23_	ν_13_
89.2	89.2	8.6	5.8	4.6	4.6	0.312	0.32	0.32
X_T_ (MPa)	X_C_ (MPa)	Y_T_ (MPa)	Y_C_ (MPa)	S_12_ (MPa)	S_23_ (MPa)	S_13_ (MPa)	Z_T_ (MPa)	Z_C_ (MPa)
745	419	745	419	83.2	83.2	83.2	69.2	69.2
**EA9394 Resin**								
E (GPa)	ν	T (MPa)	G (MPa)	S (MPa)	Shear strain at failure (%)			-
4.235	0.35	46	1461	28.9	1.66			-
Al (LY12CZ)	-	-	-	-	-			-
E (GPa)	ν	Yielding stress (MPa)			-			-
72	0.33	327			-			-

**Table 4 materials-13-02740-t004:** Degradation rules of laminate material properties.

Failure Modes	Degradation Rules of Laminate Material Properties
In-plane warp and fill direction failure in tension	E_11_ = 0.07E_11_, E_22_ = 0.07E_22_, E_33_ = 0.07E_33_, G_12_ = 0.07G_12_, G_13_ = 0.07G_13_, G_23_ = 0.07G_23_ v_12_ = 0.07v_12_, v_23_ = 0.07v_23_, v_13_ = 0.07v_13_
In-plane warp and fill direction failure in compression	E_11_ = 0.14E_11_, E_22_ = 0.14E_22_, E_33_ = 0.14E_33_, G_12_ = 0.14G_12_, G_13_ = 0.14G_13_, v_12_ = 0.14v_12_, v_23_ = 0.14v_23_, v_13_ = 0.14v_13_
In-plane shear-out failure	G_12_ = v_12_ = 0
Out-of-plane delamination	E_33_ = G_23_ = G_13_ = v_23_ = v_13_ = 0

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
