# Peer review of "Mechanical Behavior of Single Patch Composite Repaired Al Alloy Plates: Experimental and Numerical Analysis"

_materials, 2020, doi:10.3390/ma13122740_

Round 1

Reviewer 1 Report

The authors present the preparation of glass fiber reinforced polymer (GFRP) materials used to repair cracked Al plates.

Minor correction is need:  write the references like as journal instructions.

Author Response

The references have been corrected as the journal instructions. And the references are list as below:

  1. Li, G., Hedlund, S., Pang, S. S., Alaywan, W., Eggers, J. and Abadie, C. Repair of damaged RC columns using fast curing FRP composites. Compos Part B- Eng 2003, 34(3), 261-271.
  2. Sun, W., Xu, P., and Yang, Y. Development of a simplified bond model used for simulating FRP strips bonded to concrete. Compos Struct 2017, 171, 462-472.
  3. Hawileh, R.A., Musto, H.A., Abdalla, J.A. and Naser, M.Z. Finite element modeling of reinforced concrete beams externally strengthened in flexure with side-bonded FRP laminates. Compos Part B- Eng 2019, 173, 106952.
  4. Sun, F.J., Pang, S.H., Zhang, Z.W., Fu F., and Qian, K. Retrofitting Seismically Damaged Steel sections encased Concrete Composite Walls using Externally Bonded CFRP Strips. Compos Struct 2020, 236, 111927.
  5. Toutanji, H., Dempsey S. Analytical study of the use of advanced composites for repair and rehabilitation of pipelines. J Thin-Walled Struct 2001, 39(2).
  6. Mahdi, E., Eltai E. Development of cost-effective composite repair system for oil/gas pipelines. Compos Struct 2018, 202, 802-806.
  7. Rohem, N.R.F., Pacheco, L.J., Budhe, S., Banea, M.D., Sampaio, E.M., and De Barros, S.. Development and qualification of a new polymeric matrix laminated composite for pipe repair. Compos Struct 2016, 152, 737-745.
  8. Chester, R.J., Walker, K.F. and Chalkley, P.D. Adhesively bonded repairs to primary aircraft structure. Int J Adhes Adhes 1999, 19(1), 1-8.
  9. Jones, Rhys, Lorrie Molent. Application of constitutive modelling and advanced repair technology to F111C aircraft. Compos Struct 2004, 66(1-4), 145-157.
  10. Katnam, K.B, Da Silva, L.F.M. and Young, T.M. Bonded repair of composite aircraft structures: A review of scientific challenges and opportunities. Prog Aerosp Sci 2013, 61, 26-42.
  11. Truong, Viet-Hoai, Byeong-Su Kwak, Rene Roy, and Jin-Hwe Kweon. Cohesive zone method for failure analysis of scarf patch-repaired composite laminates under bending load. Compos Struct 2019, 222, 110895.
  12. Grammatikos, Sotirios A., Evangelos Z. Kordatos, Theodore E. Matikas, and Alkiviadis S. Paipetis. On the fatigue response of a bonded repaired aerospace composite using thermography. Compos Struct 2018, 188, 461-469.
  13. Heshmati, Mohsen, Reza Haghani, and Mohammad Al-Emrani. Environmental durability of adhesively bonded FRP/steel joints in civil engineering applications: state of the art. Compos Part B- Eng 2015, 81, 259-275.
  14. Chen, T., Zhao, X.L., Gu, X.L., and Xiao, Z.G. Numerical analysis on fatigue crack growth life of non-load-carrying cruciform welded joints repaired with FRP materials. Compos Part B-Eng 2014, 56, 171-177.
  15. Fujimoto, S.E., Sekine, H. Identification of crack and disbond fronts in repaired aircraft structural panels with bonded FRP composite patches. Compos Struct 2007, 77(4), 533-545.
  16. Benyahia, F., Aminallah, L., Albedah, A., Bachir Bouiadjra, B., and Achour, T. Experimental and numerical analysis of bonded composite patch repair in aluminum alloy 7075 T6. Mater Des 2015, 73, 67-73.
  17. Albedah, Abdulmohsen, Sohail MA Khan Mohammed, Bachir Bachir Bouiadjra, Bel Abbbes Bachir Bouiadjra, and Faycal Benyahia. Effect of the patch length on the effectiveness of one-sided bonded composite repair for aluminum panels. Int J Adhes Adhes 2018, 81, 83-89.
  18. Ouinas, D., Bouiadjra, B.B., Serier, B. The effects of disbonds on the stress intensity factor of aluminium panels repaired using composite materials. Compos Struct 2007, 78(2), 278-284.
  19. Ahmed, Shafique, Erik T. Thostenson, Thomas Schumacher, Sagar M. Doshi, and Jennifer R. McConnell. Integration of carbon nanotube sensing skins and carbon fiber composites for monitoring and structural repair of fatigue cracked metal structures. Compos Struct 2018, 203, 182-192.
  20. Yen, Chian-Fong. Ballistic impact modeling of composite materials. Proceedings of the 7th international LS-DYNA users conference 2002, 6, 15-23.

Reviewer 2 Report

The paper is relatively well written and the research looks interesting. While the authors are commended on the work, in the reviewers humble opinion, much more information needs to be provided for this article to be published in a journal.

1.       The literature review is almost non-existent. There are numerous studies that deal primarily with FRP strengthening of Aluminum beams, tubes, columns, plates etc. These need to be highlighted, with any studies looking at the FEM modeling highlighted.

2.       The mechanical properties of the Aluminum nor the GFRP are provided. While some properties are given in Table 2 under the FEA portion, it is not stated if these are test properties or assumed properties.

3.       The GFRP lengths, along with the overlap lengths in the two configurations (triangular and inverted triangular) are not provided. What was the thickness of the Aluminum plate and the overall thickness of the 6 plys?

4.       I’m assuming SW50 is the GFRP as it is not stated anywhere. Based on the properties provided in Table 2, it looks like a bi-axial fabric. The fiber orientation needs to be detailed and why a bi-axial fabric was chosen when the applied load is uniaxial needs to be explained.

5.       Why were three different resins selected? Do they have very different properties? No information on the resins are provided.

6.       While SW50 was the FRP used in the testing, Table 2 provides data for SW90-a. What is the reason for the difference?

7.       How the GFRP patch length was determined is also important. Was the length of the shortest pre-preg  greater than the development length of the bond? 

8.       Based on Figure 2, it appears that the GFRP was applied only on one side. If this is the case, the loading would not be symmetric. This needs to be clarified.

9.       Based on Figure 11, it appears to the reviewer that the GFRP patch was only applied to one side of the Aluminum plate. If so, it is important to state the depth of the crack or was it a through crack going all the way through the plate?  

10.   Why isn’t the load-displacement for the Aluminum plate without the crack provided? This is important as it is not clear if the load-deformation characteristics (axial stiffness) returns to the original capacity.

11.   What is the reasoning behind the statement that having 80% of the original capacity represents an excellent repair efficiency? How can something (in this case say a naval air craft) operate knowing that the structure is only at 80% capacity when compared to the original structure?  

12.   What is the tensile strength provided in Table 1? Is this the strength in the GFRP only? Or is this the strength of the total specimen? Either way, the calculation needs to be explained.

13.   What is the highlighted box in Figure 6, away from the fracture line that is called the fracture surface?

14.   It is not clear what Figure 7 is. If this is the highlighted box in Figure 6, how is this a ‘fracture surface’ when the fracture happened away from this area? If there is a difference in the surface profile after fracture, suggest showing the surface profile of the section before rupture for comparison.

15.   Was debonding at the GFRP-Aluminum surface considered in the FEA? If so, how was the bond-slip modeled?

16.   Was inter-lamina shear between the different GFRP plys considered in the analysis?

17.   In Figure 10, why are the FEA strain readings stepped? It looks like the loading was applied in steps, but this is not seen in the deflections (Fig. 9).

18.   It is not clear to the reviewer what is happening in Figure 13. Are all three failures happening one after the other? Is the delamination between the Aluminum and GFRP or within the GFRP?

A more detailed and clear explanation of the experimental and FEA process needs to be provided if the article is to be published as a journal publication. Some work on the grammar is also needed. For example:

Line 29-30- Not sure what ‘From the above research work, it was worth noting that there will be prosperity and development in FRP composite repair techniques.’ Means.

Line 46- I believe ‘agent’ should be ‘urgent’

Author Response

First of all, we appreciate your comments very much. And the authors want to thank you for the suggestions. We have revised the article according to your comments. And the responses are list as below.

Q1:The literature review is almost non-existent. There are numerous studies that deal primarily with FRP strengthening of Aluminum beams, tubes, columns, plates etc. These need to be highlighted, with any studies looking at the FEM modeling highlighted.

A1:Five more articles were reviewed in the Introduction part.

Q2: The mechanical properties of the Aluminum nor the GFRP are provided. While some properties are given in Table 2 under the FEA portion, it is not stated if these are test properties or assumed properties.

A2: The mechanical properties of the Al plate were added in Table 2. Mechanical properties of the Al plate, which is very common, were from the design manual. Mechanical properties of the GFRP laminate were from tests results.

Q3: The GFRP lengths, along with the overlap lengths in the two configurations (triangular and inverted triangular) are not provided. What was the thickness of the Aluminum plate and the overall thickness of the 6 plys?

A3: The lengths of GFRP patch laminates were 62mm,72mm,82mm,92mm,102mm,112mm each from bottom to the top for the triangular patch. And it is upside down for the inverted triangular patch. The thickness of the Al plate is 1.5mm, and overall thickness of 6 plys was approximately 0.6mm, with each layer of 0.1mm.

Q4: I’m assuming SW50 is the GFRP as it is not stated anywhere. Based on the properties provided in Table 2, it looks like a bi-axial fabric. The fiber orientation needs to be detailed and why a bi-axial fabric was chosen when the applied load is uniaxial needs to be explained.

A4: SW50 is mistake writing, it should be SW90-a. SW90-a is a kind of plain-weaved glass fabric, which is indeed a bi-axial fabric. The reason why we chose this kind of fabric is taht the load applied in practical structure was muti-directional and this kind of fabric can provide balanced mechanical properties in all directions. Also the fabric is convenient for hand lay-up operation. While for test research, uniaxial tests were common used and easily available. Although the test load was uniaxial, the test results can still be useful for practical application.

Q5: Why were three different resins selected? Do they have very different properties? No information on the resins are provided.
A5: Three different resins were selected to compare their characteristics in practical repair process and repair efficiency, so as to choose the most easily repair process. The properties of the resins were added in Table 1.

Q6: While SW50 was the FRP used in the testing, Table 2 provides data for SW90-a. What is the reason for the difference?

A6: It is the same with Q4. SW50 is mistake writing,it should be SW90-a.

Q7: How the GFRP patch length was determined is also important. Was the length of the shortest pre-preg greater than the development length of the bond?

A7: The GFRP patch length was determined based on the combine consideration of past research work and literatures. In this research, the shortest pre-preg was selected to reduce the stress concentration around the crack tips and slow the crack propagation.

Q8: Based on Figure 2, it appears that the GFRP was applied only on one side. If this is the case, the loading would not be symmetric. This needs to be clarified.

A8: Indeed, the loading in the specimen was not symmetric and bending deflection can be seen in both test results and FEM results.

Q9:Based on Figure 11, it appears to the reviewer that the GFRP patch was only applied to one side of the Aluminum plate. If so, it is important to state the depth of the crack or was it a through crack going all the way through the plate?

A9: The crack was a through crack going all the way through the plate. And the patch was single-sided.

Q10: Why isn’t the load-displacement for the Aluminum plate without the crack provided? This is important as it is not clear if the load-deformation characteristics (axial stiffness) returns to the original capacity.

A10: The load-displacement of the Al plate without crack was added in Fig 9.

Q11: What is the reasoning behind the statement that having 80% of the original capacity represents an excellent repair efficiency? How can something (in this case say a naval air craft) operate knowing that the structure is only at 80% capacity when compared to the original structure?

A11: From our engineering experience, recovery ratio, higher than 80% of the original capacity, was excellent in repaif effiency and can be applied in some structures of naval aircraft.

Q12: What is the tensile strength provided in Table 1? Is this the strength in the GFRP only? Or is this the strength of the total specimen? Either way, the calculation needs to be explained.

A12: Tensile strength in Table 1 was caculated by dividing the ultimate tensile load by cross section area.

Q13: What is the highlighted box in Figure 6, away from the fracture line that is called the fracture surface?

A13: The highlighted box in Figure 6 was the area observed by scanning electronic micrograph, as shown in Figure 7.

Q14: It is not clear what Figure 7 is. If this is the highlighted box in Figure 6, how is this a ‘fracture surface’ when the fracture happened away from this area? If there is a difference in the surface profile after fracture, suggest showing the surface profile of the section before rupture for comparison.

A14: Both Figure 6 and Figure 7 demonstrates the fracture surface of the specimen. Figure 7 was an enlarged picture of the highlighted box in Figure 6 by using scanning electronic micrograph.

Q15: Was debonding at the GFRP-Aluminum surface considered in the FEA? If so, how was the bond-slip modeled?

A15: The FEA model doesn't consider the debonding at the GFRP-Aluminum surface yet. In future study, we will try the cohesive element to model the debonding behavior.

Q16: Was inter-lamina shear between the different GFRP plys considered in the analysis?

A16: Shear failure was considered in the analysis, as shown in formula 10.

Q17: In Figure 10, why are the FEA strain readings stepped? It looks like the loading was applied in steps, but this is not seen in the deflections (Fig. 9).

A17: There was a mistake when process the strain data. Figure 10 was updated with the correct strain data from FEA analysis. And the strain positions were added by attaching a picture.

Q18: It is not clear to the reviewer what is happening in Figure 13. Are all three failures happening one after the other? Is the delamination between the Aluminum and GFRP or within the GFRP?

A18: All three failures happen one after the other. Figure 13 shows the development of the three failure. And the picture was a view of the GFRP layer adjecent to the Al plate.

Q19: Line 29-30- Not sure what ‘From the above research work, it was worth noting that there will be prosperity and development in FRP composite repair techniques.’ Means.

Line 46- I believe ‘agent’ should be ‘urgent’

A19: These grammer errors were corrected.

Reviewer 3 Report

The paper presents the laboratory and numerical experiments on the repaired Al plates with three types of repair material and two types of layers application (6 models generally). The research seems to be rather useful for practical purposes in the aircraft and civil engineering industries to repaire the cracks. In general the paper looks well-structured and coincides with engineering logic, but there are still some recommendations for corrections. Checking grammar is required: there are many grammar mistakes (as "results shows", "...was established to analysisze..", "are needed", etc.). Literature citing should be arranged in a proper way(probably [8-9] instead of [8,9] or 7 is missing in the beginning?). It would be good to add some comments for Fig.7, and to give the explanation (prediction) for the differences in laboratory and numerical experiments in Fig. 9.

Author Response

Thank you very much for your comments. We have revised the article accordingly. And the response to your comments are list as below:

The grammer was checked again and mistakes were corrected. Some comments were added for Fig.7. And the differences in laboratory and numerical experiments in Fig. 9 were explained "The boundary condition and the material failure model the FEA model can hardly be the same with those of the practical tests. It may leads to the differences between the FEA results and test results."

Round 2

Reviewer 2 Report

Thank you for addressing most of my concerns regarding the manuscript. It looks very good. Please consider making the following changes before final submission:

Line 54: 'Ouinas D et al.', should be 'Ouinas et al.'

Line 82: Please consider identifying what 'LY12CZ' is for readers who are unfamiliar with Aluminum plate types, or delete is all together if it doesn't need to be there.

Figure 1: Please consider increasing the font size and overall quality/resolution of the figure.

Table 2: Consider reorganizing the layout of the table as both the cracked plate and original plate seem to have a triangle patch (the strengths are listed under that heading). Also, calling the cracked plate failure load ratio (54.73%) as a 'recovery ratio' is not very accurate. 

Author Response

Thank you very much for your comments. We have revised according to each comment in the article. And the responses are listed as below:

Q1: Line 54: 'Ouinas D et al.', should be 'Ouinas et al.'

A1: It has been corrected as 'Ouinas et al.' in Line54.

Q2:Line 82: Please consider identifying what 'LY12CZ' is for readers who are unfamiliar with Aluminum plate types, or delete is all together if it doesn't need to be there.

A2: The Alloy LY12CZ is a common aeronautical engineering aluminum alloy. It was detailed with information in Line 83-84.

Q3: Please consider increasing the font size and overall quality/resolution of the figure.

A3: Figure 1 was replaced with a high resolution picture and the font size was enlarged, see in Line 94-96.

Q4: Consider reorganizing the layout of the table as both the cracked plate and original plate seem to have a triangle patch (the strengths are listed under that heading). Also, calling the cracked plate failure load ratio (54.73%) as a 'recovery ratio' is not very accurate.

A4: Table 2 was reorganized, the data of the cracked plate and original plate was deleted. And a note was added under the table to illustrate the original strength and residual load capcaity of the cracked Al plate, see Line 127-129. The 54.73% for the cracked plate was called as residual load capacity ratio.